# Enhancing Multi-Objective Offline RL with Adaptive Preference Integration

## Abstract

Multi-objective reinforcement learning (MORL) is crucial for real-world applications where multiple conflicting goals must be optimized, such as in healthcare or autonomous systems. Offline MORL extends these benefits by using pre-collected datasets, allowing for effective learning without continuous interaction with the environment. However, existing offline MORL algorithms often struggle with scaling across large preference spaces and handling unknown preferences during evaluation. To address these challenges, we propose the Preference-Attended Multi-Objective Decision Transformer (PA-MODT), a novel architecture that integrates a preference-attention block with a modular transformer structure. This design enables effective generalization over different preferences and trajectories, providing a more robust approach to generating optimal Pareto fronts. We tested PA-MODT on five D4MORL datasets with millions of trajectories representing various objectives and found that it consistently outperforms existing models, achieving Pareto fronts that align closely with behavioral policy. This demonstrates PA-MODT's potential to effectively manage complex multi-objective reinforcement learning tasks.

## 1 Introduction

Offline reinforcement learning (OfflineRL) has seen a surge in popularity due to its capability to effectively utilize pre-existing datasets to train optimal policies without the need for direct interaction with the environment. This approach offers a data-driven pathway to learning, allowing for sample-efficient policy optimization Prudencio et al. (2023). After training, the policies can either be fine-tuned through environmental interactions or deployed for immediate use, making OfflineRL a versatile tool in applications where real-time interaction is expensive or risky, such as autonomous driving, robotics manipulation, and dialog generation Levine et al. (2020). Various OfflineRL algorithms have emerged, encompassing both model-based approaches from Kidambi et al. (2020); Yu et al. (2020; 2021), and model-free strategies from Fujimoto & Gu (2021); Kumar et al. (2019); Wu et al. (2019), typically leveraging temporal difference learning Sutton & Barto (2018) or value function estimation. An alternative route has also been explored with Decision Transformer Chen et al. (2021) and Reinforcement Learning via Supervised Learning (RvS) Emmons et al. (2021), which rely on autoregressive generative modeling. This shift towards OfflineRL is transforming the landscape of reinforcement learning by reducing the need for costly environmental interactions while offering flexibility for real-world applications.

In RL, the primary goal is to derive a policy that maximizes the return for a specific objective function. However, many real-world applications demand the optimization of multiple objectives simultaneously. Consider wind turbine control Hayes et al. (2022), which must balance power output with reducing component fatigue to extend turbine lifespan, or medical treatment, which involves optimizing effectiveness while minimizing side effects. Multi-objective reinforcement learning (MORL) addresses such scenarios by accommodating conflicting goals within a single framework. MORL algorithms in online settings generally focus on predefined preferences, either targeting a single optimal policy Van Moffaert et al. (2013); Roijers et al. (2013) or generating multiple policies Mossalam et al. (2016); Roijers et al. (2014) to cover a range of desired outcomes. This versatility makes MORL a robust approach for complex, real-world optimization problems where balancing multiple conflicting objectives is essential.

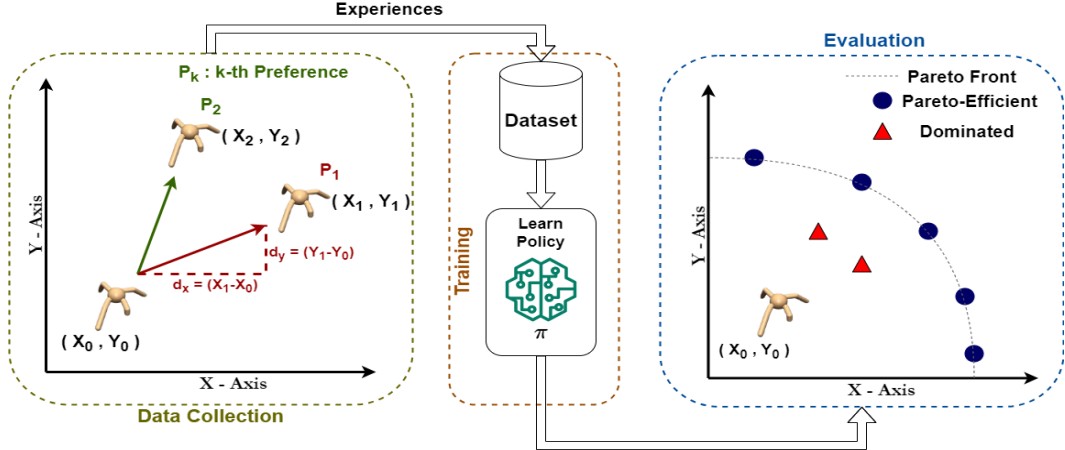

Figure 1: Overview of MORL process using D4MORL's MO-Ant dataset.
The MORL process, as demonstrated by D4MORL's MO-Ant dataset, involves assigning a 2-dimensional reward for movement in each direction, where one dimension corresponds to the x-axis and the other to the y-axis, i.e., $[r_x \propto d_x, r_y \propto d_y]$. The agent's overall direction of motion is influenced by the preference given to each direction. Experiences collected across various preferences form an offline MORL dataset (Data Collection Phase). Subsequently, a model is trained using MORL algorithms to formulate an optimal policy (Training Phase). This policy is evaluated by generating Pareto front corresponding to observed and unobserved preferences in the dataset (Evaluation Phase). The Pareto front ultimately represents the ant's movement for various preferences, highlighting the trade-offs and optimal solutions achieved.

Offline MORL extends the advantages of OfflineRL by using previously collected datasets to optimize learning across multiple objectives, eliminating the need for real-time interaction with the environment. An overview of the offline MORL process is shown in Figure 1. Similar to online MORL, offline MORL algorithms Wu et al. (2021); Thomas et al. (2021) aims to create either a single optimal policy or an ensemble of policies based on predefined target preferences. However, offline MORL often struggles with scalability across a large preference space [1] and managing unknown preferences a priori. To address these issues, the Pareto Efficient Decision Agent (PEDA) framework Zhu et al. (2023) integrates preference information into conventional OfflineRL inputs (i.e., states, actions, reward), creating preference-conditioned trajectories by concatenating preferences with other inputs. This approach allows policies to generalize across both trajectories and preferences. The PEDA framework introduced large-scale datasets derived from the MuJoCo environment Xu et al. (2020), containing millions of pre-recorded trajectories, demonstrating effective methods for building multi-objective decision transformers and multi-objective RvS models to tackle complex MORL tasks.

Preferences, which are fixed within a single trajectory example in a dataset, are time-independent features. In contrast, states, actions, and returns are time-dependent features that vary with each time step. MODT uses a transformer architecture Vaswani et al. (2017) to predict future actions by training on a mix of these time-dependent and time-independent features. Prior studies indicate that simply adding preferences as an extra token along with time-dependent features in the transformer can create a weak correlation between preferences and predictions Zhu et al. (2023). However, concatenating these two types of features has proven to be a more effective method for autoregressive training, known as preference conditioning. On the other hand, the authors in Ghanem et al. (2023) point out that the decision transformer in online settings Zheng et al. (2022) may not fully leverage the transformer model's potential for future action prediction, as distinct attention blocks tend to learn uniform patterns. This challenge indicates that integrating preferences directly with trajectory-based time-dependent features could result in inefficient model utilization and limit the flexibility of preference prediction. By contrast, modifications to transformer architecture targeting specific

---

[1]Preference space refers to potential preferences or choices, where each point represents a unique combination of objectives or trade-offs. Preferences for individual objectives can vary between 0 and 1, indicating their relative importance.

problem statements have yielded improved outcomes Li et al. (2022); Cai & Rostami (2024); Kim et al. (2021); Yu et al. (2023).

These insights prompted the development of a new transformer-based architecture: the Preference-Attended Multi-Objective Decision Transformer (PA-MODT). This architecture is specifically designed to handle MORL tasks by incorporating a unique preference-attention block within a modular transformer structure. PA-MODT is effective at generalizing across the preference space and has been shown to outperform existing models, including MORvS. This paper also delves into the sensitivity of evaluation metrics to the derived Pareto front, providing a comprehensive understanding of the model's performance in various MORL scenarios. Furthermore, we provide Pareto front visualizations obtained using the PA-MODT model. Our findings underscore that even slight changes in evaluation metrics values can lead to significant variations in the resulting Pareto fronts, ultimately influencing model assessments and selections.

## 2 RELATED WORK

**Offline RL** The key challenge of OfflineRL is handling the out-of-distribution behavior, referred to as distribution shift Levine et al. (2020); Prudencio et al. (2023). A class of algorithms applies behavior policy regularization to avoid distribution shift so that the learned policy stays close to the behavior policy. Batch Constraint Deep Q-learning (BCQ) Fujimoto et al. (2019) applied off-policy learning, combining the Q-network with a state-conditioned variational auto-encoder to model the behavior policy distribution. Bootstrapping Error Accumulation Reduction (BEAR) Kumar et al. (2019) is an actor-critic algorithm that uses maximum mean discrepancy between samples from the learned policy and pre-modeled behavior policy as a policy regularization method. Behavior regularized actor-critic (BRAC) Wu et al. (2019) introduced a general policy regularization framework by evaluating previous works extensively.

Some algorithms learn a conservative Q-function by learning a lower bound of the true value function to handle distribution shifts. Conservative Q-learning (CQL) Kumar et al. (2020) and Conservative Offline Model-Based Policy Optimization (COMBO) Yu et al. (2021) prevent the overestimation of value function due to out-of-distribution actions using the above method. A few algorithms do not explicitly handle distribution shift by applying any restriction but still handle the issue using the single step of policy evaluation and improvement Kostrikov et al. (2021); Brandfonbrener et al. (2021), i.e., without off-policy evaluation.

**MORL** In online settings, a group of MORL strategies involves training a single policy corresponding to a single preference vector by transforming the MORL problem into a single objective RL problem through techniques such as scalarization Agarwal et al. (2022), combining objectives in distributional space Abdolmaleki et al. (2020). A few single policy algorithms Abels et al. (2019); Basaklar et al. (2022); Yang et al. (2019) aim to approximate the Pareto front with a single policy that generalizes over preference space and resolves the scalability issues in previously stated methods. The other group of strategies involves obtaining an ensemble of multiple policies. A method of obtaining multiple policies is to apply single policy algorithms for multiple preferences Mossalam et al. (2016); Roijers et al. (2014). Xu et al. (2020) uses a prediction-guided evolutionary learning algorithm to obtain a set of disjoint policies corresponding to different segments in the Pareto front space. Handa (2009) extends the Estimation of Distribution Algorithms approach to estimate multiple policies in the MORL problem.

**Offline MORL** MORL in offline settings has gotten attention recently, and a limited number of works exist in this domain. Pessimistic Dual Iteration (PEDI) Wu et al. (2021) employs dual gradient descent with pessimism while formulating the constraint problem (non-interactions with the environment) as a primal-dual problem to find an optimal policy for a fixed preference vector. Thomas et al. (2021) extends the work of Laroche et al. (2019) in multi-objective settings by adapting the Seldonian framework for safe policy improvement for predefined preferences. Zhu et al. (2023) utilizes decision models to find a single policy for all preferences by providing the information as input to the models they evaluated on the proposed dataset D4MORL. Lin et al. (2024) extends existing offline policy regularization method for single objectives into multi-objective settings, which is then evaluated on the D4MORL and proposed MOSB datasets.

## 3 PRELIMINARIES

**Setup and Notation**   A MORL environment is generally formulated as multiobjective markov decision process, represented by the tuple $< S, A, P, R, \gamma, F, \Omega >$. The MOMDP tuple consists of states ($s \in S$), actions ($a \in A$), a transition distribution $P(s'|s, a)$, a reward function ($R$), a discount factor ($\gamma$), a preference-reward mapping function ($F$), and preferences ($p \in \Omega$). At any timestep $t$, the next state of the agent is obtained using the transition function: $s_{t+t} \sim P(s_{t+1}|s_t, a_t)$. The reward function generates the vector reward based on state and action as $r = R(s, a) = [R_1(s, a), R_2(s, a), ..., R_n(s, a)]$, where $n$ is the total number of objectives and $R_i(s, a)$ is the reward obtained for $i^{th}$ objective. A trajectory contains the transitions taken by the agent, which is represented as $(s_0, a_0, r_0, s_1, a_1, r_1, ..., s_T, a_T, r_T)$ where $T$ denotes the length of the trajectory. The vector-valued return at any timestep $t$ in a trajectory is the discounted sum of the reward obtained till the current timestep, given by $\Gamma_t = \sum_{i=0}^{t} \gamma^i \cdot r_i$. On the other hand, the return-to-go at any timestep $t$ represents the future return of the trajectory from the current timestep, given by $g_t = \sum_{i=t}^{T} \gamma^{i-t} \cdot r_i$. A preference-reward mapping function maps a vector-valued reward to a scalar utility value. In this paper, we have used a linear preference-reward mapping function i.e. $F(r, p) = r \cdot p^T$, where p is a preference vector such that $p \in \Omega$.

**Decision Transformer**   A family of OfflineRL algorithms tries to find an optimal policy that maximizes the expected return on any given state in an MDP by applying supervised learning on some prerecorded RL dataset $D = \{(s, a, s', r)\}$. Decision Transformer (DT) is an OfflineRL algorithm that formulates the RL problem as a sequence modeling task. DT applies autoregressive training and generation on the GPT model to predict future action tokens by feeding return-to-go, states, and actions as input. One input token contains a combination of return-to-go, state, and action embeddings given by $(g, s, a)$. A total of $K$ such input tokens are used where $K$ is called the context length, and therefore, the final input is denoted as $\tau = (g_1, s_1, a_1, g_2, s_2, a_2, ..., g_K, s_K, a_K)$. For the evaluation, the desired return, along with initial state is used to generate the required/optimal trajectory.

**Pareto Optimality**   The goal of MORL is to obtain a policy ($\pi(a|s, p)$) that maximizes the expected return of the induced trajectory for any given preference vector $p$. A policy is evaluated for a set of preferences containing a finite number of preference vectors, $E = [p_1, p_2...p_m]$. A solution set of returns ($\Delta$) is constructed containing returns corresponding to each preference vector. A point $\beta$ in the feasible solution set is considered Pareto efficient if there exists no other point $\alpha$ in the feasible solution set such that $R_i^{\beta} < R_i^{\alpha}$ for at least one objective function $i \in \{1, 2, ..., n\}$, where $R_i^{\beta}$ and $R_i^{\alpha}$ denote the values of objective function $i$ at points $\beta$ and $\alpha$, respectively. Mathematically, this can be expressed as: $\nexists \alpha \in \Delta, \alpha \neq \beta : R_i^{\beta} < R_i^{\alpha}, \exists i \in \{1, 2, ..., n\}$. The curve traced by all Pareto-efficient points is called the Pareto front, as shown in Figure 1. The goodness of the policy is measured by evaluating the obtained Pareto front on Hypervolume and Sparsity matrices.

## 4 ARCHITECTURE

MORL poses significant challenges, as it requires balancing time-dependent features like states, actions, and return-to-go with time-independent features such as preferences. To achieve robust generalization and effective decision-making in complex environments, finding an optimal architecture that integrates these diverse features is crucial. We address this challenge by focusing on transformer-based models for their capacity to handle complex data and explore various approaches to effectively integrate preference-based information. We experiment with different architectural configurations to determine the most effective strategy for multi-objective RL:

- **PA-MODT** (Preference-Attended Multi-Objective Decision Transformer): A transformer-based configuration featuring a Preference-Attention (PA) block and dedicated preference-based input encoding.

- **Experiment D** (Direct Input to the PA Block): Preferences are directly fed into the PA block of the PA-MODT configuration without passing through a preference embedding layer.

- **Experiment F** (Feed-Forward Layer After the PA Block): A feed-forward layer follows the PA block in the PA-MODT configuration, mirroring the structure of traditional transformer models where attention layers are succeeded by a feed-forward layer.

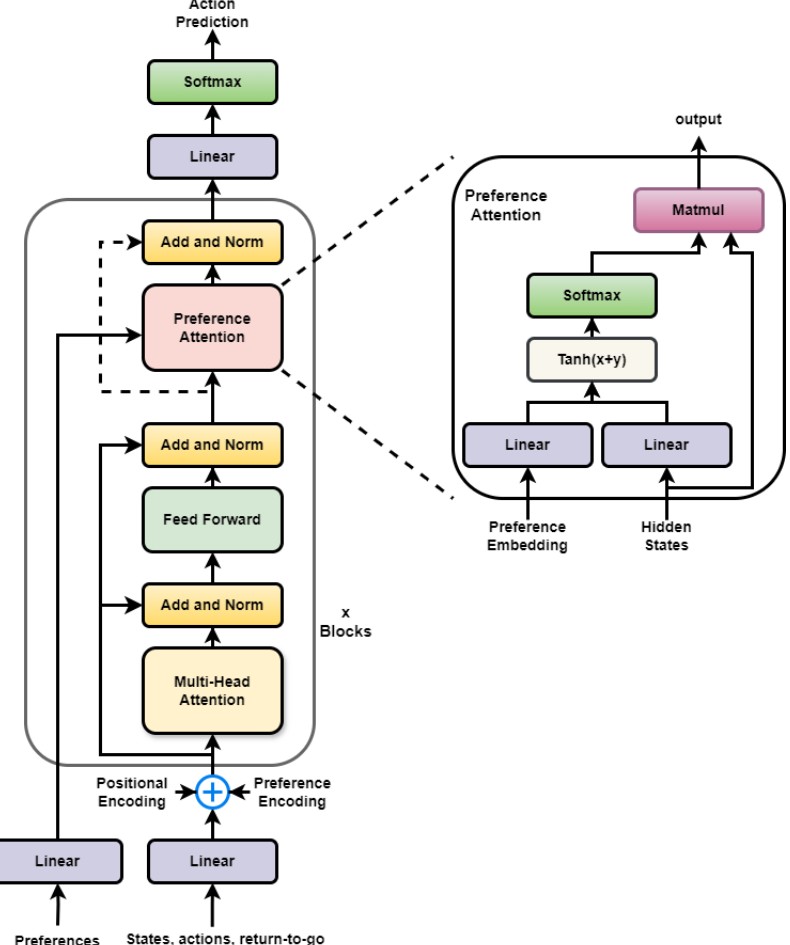

Figure 2: Preference-Attended Multi-objective Decision Transformer Architecture

PA-MODT demonstrates superior performance in MORL tasks. Experiment D shows reduced performance compared to PA-MODT, highlighting the significance of preference embeddings. Conversely, Experiment F performs comparably to PA-MODT, suggesting that the additional feed-forward layer may not offer substantial benefits and could be considered redundant. These results are presented in Section A.1. We identify the key features contributing to PA-MODT's success in MORL upon conclusion of these experiments:

1. **Preference-Based Input Encoding**: A separate preference encoding accompanies the standard positional encoding, enabling the model to consider preferences akin to time-based positional encoding. This addition enables the architecture to take into account a broader range of preference-based information without affecting the core transformer structure.

2. **Separate Preference-Attention Pathway**: The PA block is a dedicated pathway for handling preferences, allowing the model to focus on preferences independently from other time-dependent features like states and actions. This structure offers a more dynamic approach to handling varying preferences in complex RL environments.

3. **Modular Transformer Architecture**: PA-MODT retains the fundamental components of a traditional transformer model, providing a stable and familiar foundation while allowing for flexible integration of preference-based attention through the PA block.

As illustrated in Figure 2, preference embeddings are derived through two separate linear layers: one set is directed to the PA block for preference-based attention, while the other set is used for preference-based input encoding. Additional embeddings for positional or time-step, state, action, and return-to-go are generated using dedicated linear layers. This structure ensures a clear separation between preference-based data and other time-dependent data, promoting efficient processing and decision-making in MORL tasks.

**Preference-Attention Block**    The PA block in our PA-MODT architecture processes hidden states derived from the time-dependent self-attention layer, followed by a feed-forward computational layer. It also integrates preference embeddings to generate attention scores through either an additive or a multiplicative attention mechanism. These scores determine the weights applied to the hidden states, as specified in Equations 1 - 5. The adjusted hidden states are then passed through a layer normalization step and combined with the original hidden states via a residual connection. This process creates a robust pathway for handling preferences within the model.

$$H = linear(hidden\_states) \tag{1}$$
$$\rho = linear(preference\_embeddings) \tag{2}$$
$$scores = softmax(tanh(H + \rho + Bias)) \tag{3}$$
$$\overline{hidden\_states} = hidden\_states \times scores^T \tag{4}$$
$$hidden\_states = hidden\_states + layernorm(\overline{hidden\_states}) \tag{5}$$

The PA-MODT architecture effectively integrates preference-based attention, contributing to improved performance and generalization in MORL tasks. The model's training process closely resembles that of a Decision Transformer, ensuring a straightforward implementation. A minibatch of transitions with context length $K$ is sampled from the offline RL dataset. The model uses this context to predict future actions, and the predictions are then compared against actual actions to compute a loss using the mean-squared loss function. The detailed training steps are outlined in Algorithm 1, which is provided in Appendix A.3.

## 5 EXPERIMENTS AND RESULTS

The outline of our experiments section is designed to answer the following questions: 1. How does our PA-MODT model's performance compare to that of existing models? 2. What is the effect of each component on the performance of the overall PA-MODT model? 3. How does the Pareto front shift with slight variations in each evaluation metric?

Additionally, we include a detailed Pareto front visualization in Section A.2, along with a comparison study. This visual representation helps clarify how PA-MODT performs, adding depth to our experimental results.

### 5.1 EVALUATIONS ON OFFLINE MORL BENCHMARKS

To evaluate the performance of the PA-MODT architecture, we focus on two key metrics for Pareto front analysis: Hypervolume and Sparsity. These metrics offer a comprehensive understanding of the Pareto fronts generated during the evaluation phase after model training. Hypervolume measures the area encompassed by all points on the Pareto front, indicating the extent to which the curve spreads outward from the origin. Sparsity, on the other hand, assesses the average distance between consecutive points, providing an indication of the curve's density. This graphical representation of Hypervolume assumes an optimization goal of maximizing the reward for each objective.

We conduct a comparative analysis of the performance of the PA-MODT model against existing MODT and MORvS models, which are utilized for multi-objective optimization in offline paradigms. The MODT and MORvS models utilize states, actions, and return-to-go concatenated with preference information, as outlined in the PEDA paper Zhu et al. (2023). The authors of the PEDA paper previously demonstrated that MODT and MORvS outperformed other multi-objective optimization algorithms, such as MO-CQL, MO-IQL, and Behavior Cloning (BC), in both their preference-conditioned and non-preference-conditioned variants. It is important to note that operational differences in our experiments may result in slight discrepancies in MODT and MORvS

Table 1: Results on D4MORL Amateur and Expert datasets. B indicates the performance of behavior policies from PEDA. PA-MODT is compared against preference-conditioned MODT and MORvS.

| | Dataset | Metric | B | MODT | MORvS | PA-MODT |
|---|---|---|---|---|---|---|
| Expert | Ant | HV ($10^6$) | 6.32 | $6.183 \pm 0.125$ | $6.376 \pm 0.012$ | $\mathbf{6.410 \pm 0.018}$ |
| | | SP ($10^4$) | - | $\mathbf{0.724 \pm 0.121}$ | $0.973 \pm 0.238$ | $\mathbf{0.733 \pm 0.061}$ |
| | HalfCheetah | HV ($10^6$) | 5.79 | $5.735 \pm 0.009$ | $5.757 \pm 0.012$ | $\mathbf{5.785 \pm 0.002}$ |
| | | SP ($10^3$) | - | $1.336 \pm 0.137$ | $1.396 \pm 0.487$ | $\mathbf{0.543 \pm 0.067}$ |
| | Hopper | HV ($10^7$) | 2.09 | $2.004 \pm 0.007$ | $1.826 \pm 0.041$ | $\mathbf{2.060 \pm 0.011}$ |
| | | SP ($10^5$) | - | $0.763 \pm 0.180$ | $\mathbf{0.406 \pm 0.188}$ | $0.473 \pm 0.342$ |
| | Swimmer | HV ($10^4$) | 3.25 | $3.216 \pm 0.002$ | $3.230 \pm 0.000$ | $\mathbf{3.244 \pm 0.001}$ |
| | | SP ($1$) | - | $4.486 \pm 1.048$ | $6.600 \pm 0.775$ | $\mathbf{3.363 \pm 0.096}$ |
| | Walker2d | HV ($10^6$) | 5.21 | $5.009 \pm 0.004$ | $5.006 \pm 0.063$ | $\mathbf{5.152 \pm 0.008}$ |
| | | SP ($10^4$) | - | $0.892 \pm 0.078$ | $0.649 \pm 0.148$ | $\mathbf{0.286 \pm 0.055}$ |
| Amateur | Ant | HV ($10^6$) | 5.61 | $5.982 \pm 0.029$ | $6.053 \pm 0.005$ | $\mathbf{6.111 \pm 0.006}$ |
| | | SP ($10^4$) | - | $\mathbf{0.688 \pm 0.190}$ | $0.794 \pm 0.002$ | $\mathbf{0.727 \pm 0.087}$ |
| | HalfCheetah | HV ($10^6$) | 5.68 | $5.715 \pm 0.002$ | $5.766 \pm 0.000$ | $\mathbf{5.780 \pm 0.001}$ |
| | | SP ($10^3$) | - | $\mathbf{0.393 \pm 0.038}$ | $0.615 \pm 0.152$ | $0.422 \pm 0.052$ |
| | Hopper | HV ($10^7$) | 1.97 | $1.819 \pm 0.014$ | $1.729 \pm 0.038$ | $\mathbf{1.901 \pm 0.007}$ |
| | | SP ($10^5$) | - | $0.192 \pm 0.060$ | $0.212 \pm 0.210$ | $\mathbf{0.169 \pm 0.035}$ |
| | Swimmer | HV ($10^4$) | 2.11 | $1.273 \pm 0.922$ | $2.852 \pm 0.018$ | $\mathbf{2.937 \pm 0.056}$ |
| | | SP ($1$) | - | $6.720 \pm 1.193$ | $\mathbf{1.490 \pm 0.227}$ | $4.657 \pm 0.428$ |
| | Walker2d | HV ($10^6$) | 4.99 | $4.045 \pm 0.040$ | $\mathbf{4.916 \pm 0.024}$ | $\mathbf{4.921 \pm 0.019}$ |
| | | SP ($10^4$) | - | $0.919 \pm 0.106$ | $\mathbf{0.308 \pm 0.20}$ | $\mathbf{0.292 \pm 0.045}$ |

**Note that:** High Hypervolume and low sparsity are desirable.
Expert datasets are acquired by executing actions based on the optimal reference policy derived from an ensemble of policies. Conversely, amateur dataset collection entails a similar procedure to expert dataset acquisition, with actions being taken according to a predefined probability associated with the policy.

results compared to those reported in the PEDA paper, whereas the behavior policy (B) results are directly sourced from the same study. All experiments are conducted using three different seeds, with the results presented as the average of these trials along with the standard error. Specific hyperparameters and total training steps are outlined in Appendix A.4.

As per Table 1, the PA-MODT model exhibits superior performance compared to MODT and MORvS on the D4MORL expert datasets. Furthermore, in the majority of the D4MORL amateur datasets, PA-MODT outperforms both MODT and MORvS. In a few cases, such as MO-Ant (expert and amateur), MO-HalfCheetah (amateur), and MO-Swimmer (amateur), the policies derived from PA-MODT even surpass the performance of the behavior policy. A few entries in Table 1 demonstrate modest numerical improvements. However, even small numerical differences can lead to significant variations in the quality of the Pareto front, as demonstrated in Section A.2.

## 5.2 ABLATION STUDY

We conduct an ablation study on the PA-MODT model to assess the impact of its individual components on the model's performance. At first, we remove preference encoding from the input level to understand its influence on the Pareto front. Following that, we eliminate the preference attention module, essentially transforming the model into a basic GPT-2 architecture that incorporates preference-scaled vectored return-to-go ($g' = [g_1 p_1, g_2 p_2, ..., g_n p_n]$), states, and actions with a fixed context length as inputs. All experiments are performed using the same three seeds as in the previous section, with the results averaged along with the standard error.

Table 2: Ablation Study results for PA-MODT on D4MORL Amateur and Expert datasets, where each involved component was Iteratively removed to assess its impact on performance.

| | Dataset | Metric | PA-MODT | PA-MODT (-) A | PAMODT (-) A (-) B |
|---|---|---|---|---|---|
| **Expert** | Ant | HV $(10^6)$ | $6.410 \pm 0.018$ | $6.301 \pm 0.09$ | $5.825 \pm 0.167$ |
| | | SP $(10^4)$ | $0.733 \pm 0.061$ | $0.767 \pm 0.102$ | $0.561 \pm 0.078$ |
| | HalfCheetah | HV $(10^6)$ | $5.785 \pm 0.002$ | $5.778 \pm 0.002$ | $5.632 \pm 0.090$ |
| | | SP $(10^3)$ | $0.543 \pm 0.067$ | $0.484 \pm 0.108$ | $3.980 \pm 0.693$ |
| | Hopper | HV $(10^7)$ | $2.060 \pm 0.011$ | $2.039 \pm 0.019$ | $1.957 \pm 0.055$ |
| | | SP $(10^5)$ | $0.473 \pm 0.342$ | $0.257 \pm 0.048$ | $0.363 \pm 0.050$ |
| | Swimmer | HV $(10^4)$ | $3.244 \pm 0.001$ | $3.242 \pm 0.005$ | $2.801 \pm 0.035$ |
| | | SP $(1)$ | $3.363 \pm 0.096$ | $2.576 \pm 0.174$ | $31.960 \pm 15.636$ |
| | Walker2d | HV $(10^6)$ | $5.152 \pm 0.008$ | $4.928 \pm 0.019$ | $3.373 \pm 0.208$ |
| | | SP $(10^4)$ | $0.286 \pm 0.055$ | $0.392 \pm 0.214$ | $0.554 \pm 0.324$ |
| **Amateur** | Ant | HV $(10^6)$ | $6.111 \pm 0.006$ | $5.957 \pm 0.003$ | $5.405 \pm 0.205$ |
| | | SP $(10^4)$ | $0.727 \pm 0.087$ | $1.079 \pm 0.213$ | $0.614 \pm 0.256$ |
| | HalfCheetah | HV $(10^6)$ | $5.780 \pm 0.001$ | $5.745 \pm 0.007$ | $5.624 \pm 0.035$ |
| | | SP $(10^3)$ | $0.422 \pm 0.052$ | $0.701 \pm 0.186$ | $1.668 \pm 0.159$ |
| | Hopper | HV $(10^7)$ | $1.901 \pm 0.007$ | $1.867 \pm 0.037$ | $1.809 \pm 0.017$ |
| | | SP $(10^5)$ | $0.169 \pm 0.035$ | $0.241 \pm 0.106$ | $1.808 \pm 1.118$ |
| | Swimmer | HV $(10^4)$ | $2.937 \pm 0.056$ | $1.231 \pm 0.874$ | $0.591 \pm 0.035$ |
| | | SP $(1)$ | $4.657 \pm 0.428$ | $21.986 \pm 21.905$ | $0.563 \pm 0.069$ |
| | Walker2d | HV $(10^6)$ | $4.921 \pm 0.019$ | $4.704 \pm 0.202$ | $4.287 \pm 0.228$ |
| | | SP $(10^4)$ | $0.292 \pm 0.045$ | $1.799 \pm 0.875$ | $3.245 \pm 1.911$ |

A = Preference Encoding ; B = Preference Attention.

Table 2 demonstrates that preference attention significantly enhances the quality of the Pareto front across all evaluation metrics. Additionally, preference encoding improves results in all five datasets, though its impact is less pronounced compared to preference attention. Regarding the preference attention mechanism, we experiment with both additive and multiplicative attention, observing that their performance is nearly identical. The results presented in Tables 1 and 2 are derived from experiments employing additive attention.

## 5.3 PARETO FRONT SENSITIVITY TO EVALUATION MATRICES

In this subsection, we present insightful observations regarding changes in Pareto front to hypervolume and sparsity, which enhance understanding of the results presented in the previous subsections.

Figure 3 contains two Pareto front obtained for the HalfCheetah expert dataset for two different scenarios. It is visible that the Pareto front presented in Figure 3a has a few missing points, and the Pareto front in Figure 3b is dense. The difference in the hypervolume for the two figures is minimal, but the change in Pareto front is significant, which indicates a high sensitivity. On the other hand, these variations in the Pareto fronts are indicated by the considerable difference in the sparsity. For a Pareto front, the point in the middle of the curves serves an essential role in the optimization as the preference for all the objectives is nearly equal for these points, and therefore, missing a few points in that area is not desirable. Similarly, Figure 4 demonstrates two Pareto fronts obtained for the Walker2d-expert dataset for two different scenarios. The Pareto front in Figure 4b includes a few more points, as shown in the upper-left portion of the graph, as compared to the Pareto front in Figure 4a. Due to these few extra points, the sparsity of the Pareto front in Figure 4b is very high with respect to the Pareto front in Figure 4a. Although the inclusion of these few extra points is indicated by hypervolume, Pareto front's high sensitivity towards sparsity is apparent.

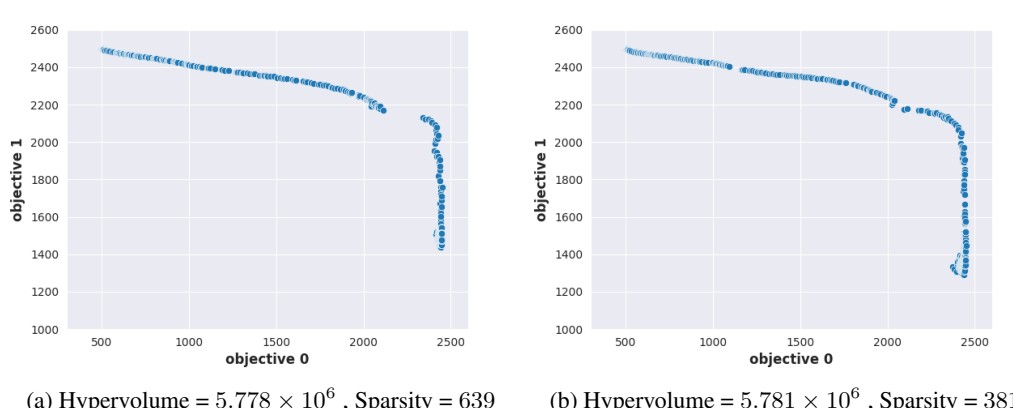

(a) Hypervolume = $5.778 \times 10^6$ , Sparsity = 639     (b) Hypervolume = $5.781 \times 10^6$ , Sparsity = 381

Figure 3: Pareto fronts for HalfCheetah-expert dataset.

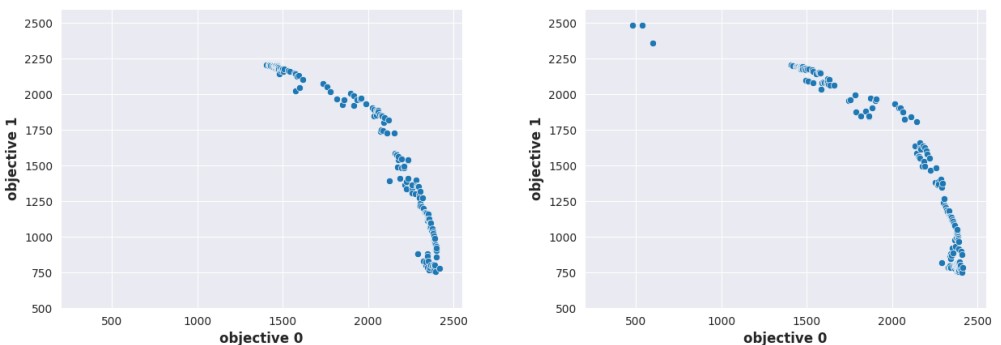

(a) Hypervolume = $4.95 \times 10^6$ , Sparsity = $0.23 \times 10^4$ (b) Hypervolume = $5.10 \times 10^6$ , Sparsity = $1.48 \times 10^4$

Figure 4: Pareto fronts for Walker2d-expert dataset.

Finally, we conclude that the Pareto front is highly sensitive to the evaluation matrics, and a comparison between two Pareto fronts should be made considering both the evaluation matrics simultaneously. It's noteworthy that even a slight change in evaluation metrics, such as those at the decimal place, can lead to a significant alteration in the Pareto front.

## 6 DISCUSSION AND FUTURE WORK

We presented an empirical analysis of how utilizing the input preferences through structural advances in transformer architecture can improve the model's performance on MORL tasks. A potential drawback of our approach is the computational complexity, as multi-objective reinforcement learning tasks are computationally intensive. Our proposed model, PAMODT, is also computationally expensive, especially when dealing with large input sequences or high-dimensional preference embeddings. This highlights the need for further research into more efficient architectures and optimization techniques that can reduce the computational burden while maintaining or enhancing performance. Additionally, we have identified a pressing need to evaluate the effectiveness of preference-attended architecture for online fine-tuning, particularly after deriving an initial policy from offline Reinforcement Learning in multi-objective optimization contexts. Throughout this paper, our focus has primarily been on predicting future actions, yet we acknowledge the potential benefits of including preferences, states, and return-to-go in these predictions. This prompts important questions regarding the applicability of preference attention beyond the offline MORL domain and the potential improvement of prediction abilities by adding future state, return-to-go, and prefer-

ence prediction. Addressing these inquiries not only enhances the reliability of preference attention but also advances the broader landscape of MORL algorithms.

## 7 REPRODUCIBILITY STATEMENT

We have used the open-source dataset and code provided by Zhu et al. (2023) for the MORL task. The algorithm used in our approach is outlined in Section A.3, the hyperparameter and experimental details are provided in Section A.4, and the computational details are mentioned in Section A.5. The modifications made to the code provided by previous authors are described in Section 4.

With the above details, our approach is easily reproducible. We would like to thank the authors Zhu et al. (2023) for providing the dataset and code, which have been instrumental in the development of our approach.

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

# A APPENDIX

## A.1 COMPARISON WITH BASELINES

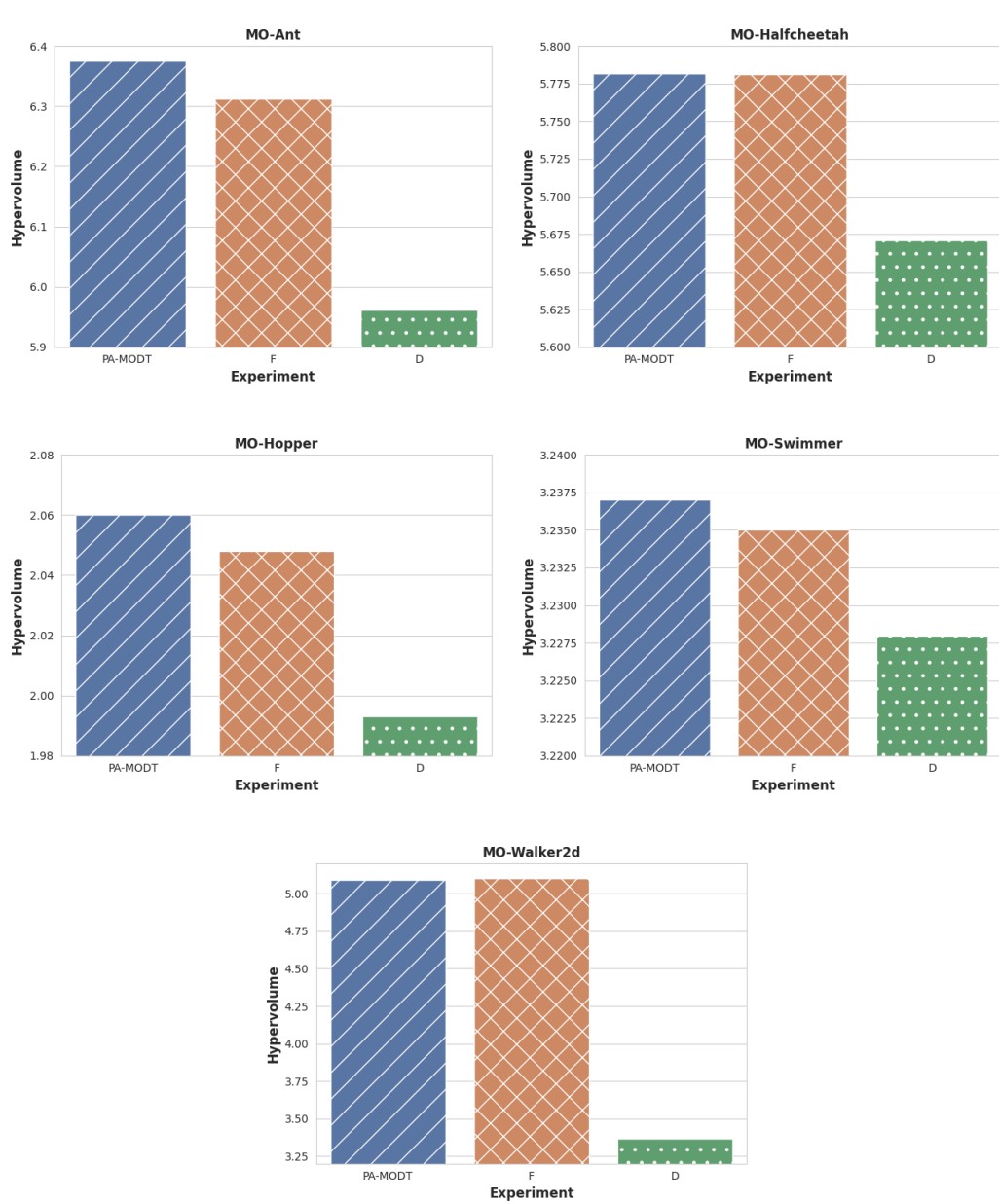

Figure 5: Comparion with baselines for D4MORL-expert datasets.

To evaluate the performance of the PA-MODT model, we conducted a comparative analysis against baseline models from Experiment D 4 and Experiment F 4. Figure 5 provides a visual representation through barplots, showcasing the hypervolumes achieved across various D4MORL-Expert datasets for each experimental setup. Notably, all experiments were carried out with the same set of hyperparameters to ensure consistency in the comparison. Since the hyperparameter search is not applied here, it is important to note that the results for PA-MODT are different from the ones presented in Table 1. This difference could be attributed to the lack of optimization of hyperparameters for PA-MODT in this specific experiment.

The results presented in Figure 5 highlight a clear pattern. Adding a feedforward layer to the PA-MODT model, as in Experiment F, tends to either yield similar performance or perform worse than the original PA-MODT architecture. This outcome suggests that additional feedforward layers may introduce unnecessary complexity, hindering the model's efficiency and adaptability. Additionally, feeding preferences directly into the transformer without dedicated embeddings, as in Experiment D, resulted in the worst performance among the tested configurations. This finding underscores the critical role that preference embeddings play in facilitating proper integration of preference-based information within the transformer architecture.

Overall, these comparative results suggest that the original PA-MODT model—with its unique Preference-Attention (PA) block and preference-based input encoding—strikes the optimal balance for MORL tasks. The performance trends observed in these experiments indicate that attempts to simplify or overly complicate the architecture can lead to diminished results, reinforcing the importance of carefully designing preference-based attention mechanisms in MORL systems.

## A.2  PARETO FRONT VISUALIZATION

Figure 6 contains the Pareto fronts obtained using the PA-MODT and MODT models on the D4MORL datasets. These visualizations are based on the best-performing seed among the three seeds discussed in Table 1. The graphs provide three key insights: the Pareto front visualization for the PA-MODT model, the comparison between the Pareto fronts of PA-MODT and MODT models, and the observation that slight changes in the metrics discussed in Table 1 result in significant variations in the Pareto fronts. For more detailed observations, it is advisable to consider the evaluation metrics values from Table 1 along with these visualizations.

Figure 6a illustrates the results for the MO-Ant dataset, where the Pareto front for PA-MODT is more widely spread compared to the MODT, which is dense in a smaller region. This indicates lower sparsity for MODT but a higher hypervolume for PA-MODT, making the PA-MODT front more desirable. Figure 6b presents the MO-HalfCheetah dataset, showing that the Pareto front for PA-MODT is denser and more spread out than for MODT. This observation is supported by the higher hypervolume and lower sparsity metrics. Figure 6c shows the MO-Hopper dataset, where the Pareto front for PA-MODT is significantly denser compared to MODT. This desirable characteristic is again reflected by the higher hypervolume and lower sparsity metrics. For the MO-Swimmer dataset, depicted in Figure 6d, the differences between the models are not visible with the current axis scale choices. Finally, Figure 6e displays the MO-Walker2d dataset results. The Pareto front for PA-MODT is broadly spread in the desired region, in contrast to the dense, smaller region of the MODT front. Due to the high concentration of Pareto-efficient points in the Pareto front of the MODT model, the sparsity is low. However, the Pareto front for PA-MODT is more diverse and spread out, which is desirable and is observed using hypervolume, making hypervolume the considered evaluation metric in this case.

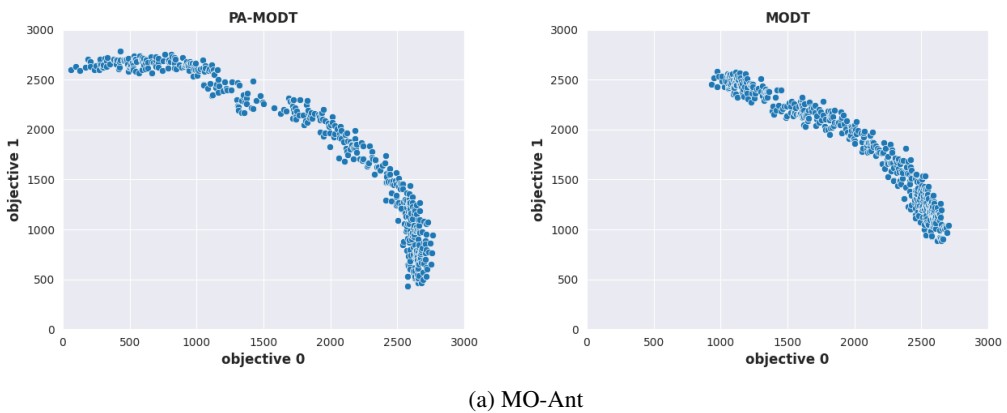

(a) MO-Ant

Figure 6: Pareto fronts for D4MORL-expert datasets.

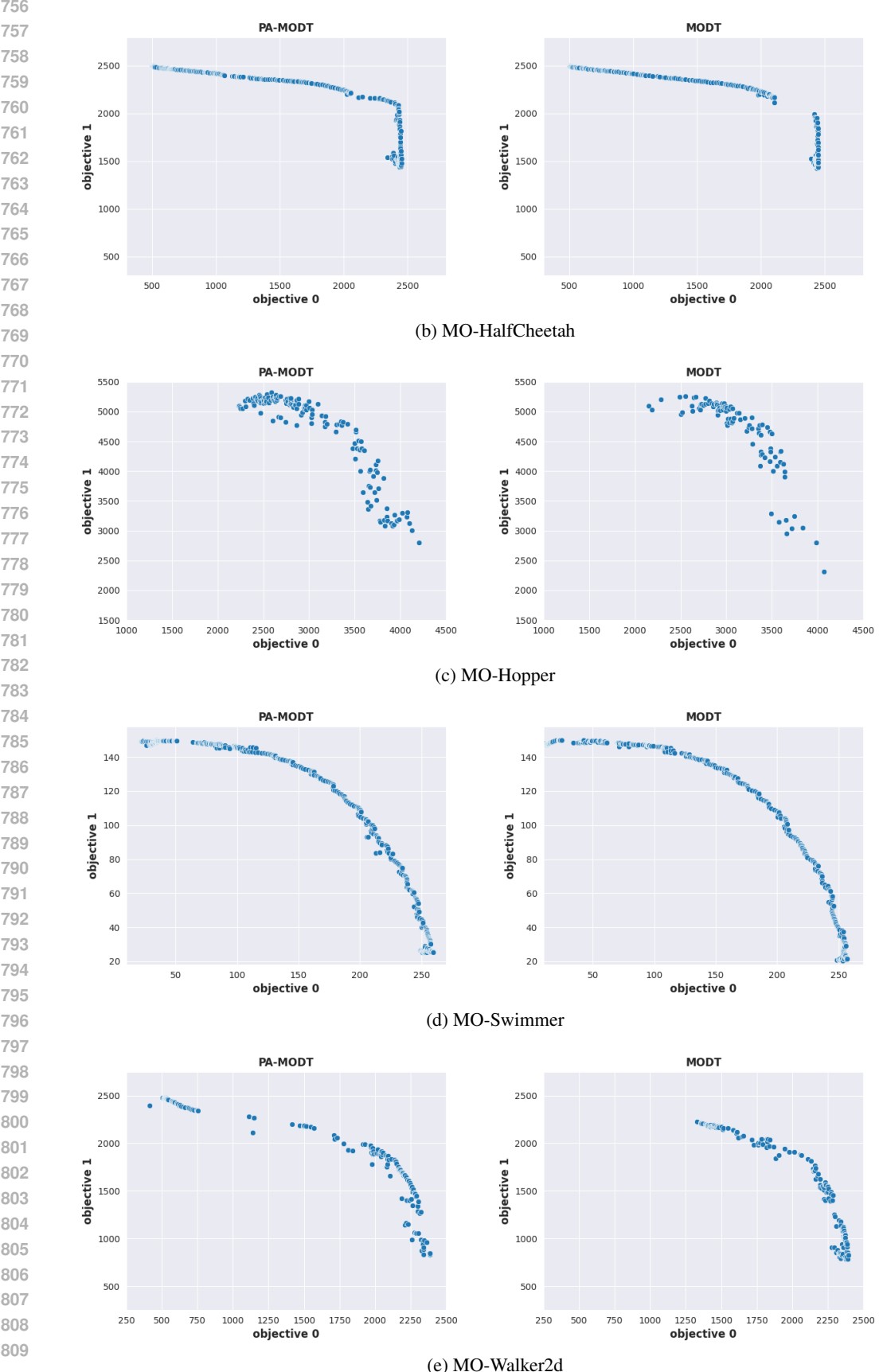

Figure 6: Pareto fronts for D4MORL-expert datasets.

Overall, these Pareto fronts in Figure 6 demonstrate the PA-MODT model's capacity to explore a range of optimal solutions and manage trade-offs among various objectives effectively. The observed robustness suggests that the model can handle complex RL scenarios where navigating multiple objectives is crucial. This visual representation underscores the model's effectiveness and adaptability in multi-objective RL tasks.

## A.3 ALGORITHM

The algorithm involves PA-MODT model and comprehensive input data preparation, including the retrieval of various embeddings. Detailed descriptions of the training and evaluation procedures, as well as the data preparation steps, are provided in the following algorithm:

---
**Algorithm 1** PAMODT Pseudocode

---
1: {# Generating predictions using PA-MODT}
2: **def MODEL(transition):**
3: {# Fetching various embeddings using dedicated linear layers}
4:    $positional\_embed = linear(transition.timestep)$
5:    $pref\_encod\_embed = linear(transition.preference)$
6:    $pref\_att\_embed = linear(transition.preference)$
7:    $state\_embed = linear(transition.state) + positional\_embed + pref\_encod\_embed$
8:    $action\_embed = linear(transition.action) + positional\_embed + pref\_encod\_embed$
9:    $rtg\_embed = linear(transition.rtg) + postional\_embed + pref\_encod\_embed$
10:    $input\_embed = stack(state\_embed, act\_embed, rtg\_embed)$
11: {# hidden states are obtained from PA-MODT using the generated embeddings}
12:    $hidden\_states = PAMODT(input\_embed = input\_embed, preference\_embed = pref\_att\_embed)$
13:    $out\_action\_embed = unstack(hidden\_states).actions$
14:    **return**   $linear(out\_action\_embed)$
15: {# returning the actions translated from hidden states using a linear layer}
16:
17: {# model training function}
18: **def train():**
19:    **for** $transition\ in\ dataset$ :
20:      $action\_Prediction = MODEL(transition)$
21:      $loss = lossFn(action\_prediction, transition.future\_action)$
22: {# backpropagate the loss}
23:
24: {# model evaluation function}
25: **def evaluate():**
26:    $target\_return = 1, state = env.reset(), action = [], done = False$
27: {# preferences is a user provided list of preferences for evaluation}
28:    **For** $p\ in\ preferences$ :
29:      **While** $not\ done$ :
30:        $action = MODEL((state, action, target\_return, p, done))$
31:        $state, reward, done = env.step(action)$
32:        $target\_return = target\_return - reward$

---

## A.4 HYPERPARAMETERS AND EXPERIMENTAL DETAILS

In this study, we have used a set of carefully designed hyperparameters that were selected through a rigorous hyperparameter search using the Optuna framework Akiba et al. (2019). This search aimed to find the optimal dataset-specific hyperparameters for each dataset.

Table 3 presents the common hyperparameters used for all experiments across all datasets. These hyperparameters were chosen based on their effectiveness and consistency across different datasets. Table 4 presents the dataset-specific hyperparameters used for each dataset in our study. While some hyperparameters were kept constant across all datasets, others were adjusted to optimize performance for each specific dataset.

Table 3: Common Hyperparameters Used in the Experiments

| Hyperparameter | Value |
|---|---|
| Optimizer | AdamW |
| Batch Size | 64 |
| n_head | 1 |
| Warmup Steps | 10K |
| Granularity (evaluation) | 500 |
| Context Length - K (evaluation) | 5 |

Table 4: Dataset-Specific Hyperparameters Used in the Experiments

| Hyperparameter | MO-Ant | MO-Walker2d | MO-Swimmer | MO-HalfCheetah | MO-Hopper |
|---|---|---|---|---|---|
| Context Length - K (training) | 22 | 24 | 20 | 24 | 22 |
| Embedding Size | 1024 | 256 | 512 | 512 | 512 |
| Blocks | 5 | 4 | 3 | 5 | 3 |
| Dropout | 0.25 | 0.15 | 0.15 | 0.2 | 0.25 |
| Learning Rate | 6.58e-5 | 3.78e-5 | 1.39e-5 | 6.21e-5 | 4.06e-5 |
| Weight Decay | 5.6e-4 | 1e-4 | 1.4e-4 | 5.6e-4 | 4.6e-4 |

The training process involves a set number of steps for each of the D4MORL datasets. Table 5 provides the total number of training steps for each dataset, along with the corresponding range. For instance, the MO-Swimmer dataset is trained between 265K and 290K steps. To evaluate the model's performance, an evaluation is conducted every 5K steps within the specified training step range for each dataset. This approach allows for monitoring the model's progress over time and helps identify the point of optimal performance.

Table 5: Total Training Steps for D4MORL Datasets

| Dataset | Total Training Steps (Range) |
|---|---|
| MO-Swimmer | 265K - 290K |
| MO-Ant | 50K - 75K |
| MO-Walker2d | 425K - 450k |
| MO-HalfCheetah | 75K - 100K |
| MO-Hopper | 450K - 475K |

## A.5 COMPUTATIONAL DETAILS

All computations in this paper were conducted on a NVIDIA A100 server equipped with 8 GPUs, each with 80GB of memory. Experiments on the D4MORL datasets, averaging 6 hours per single-seed run, utilized $30 - 40\%$ of GPU capacity, allowing two simultaneous runs per GPU core. This indicates that, given the specified hyperparameters and dataset, our experiments could feasibly be executed on servers with lower GPU configurations.

