# OpenReview forum: "Enhancing Multi-Objective Offline RL with Adaptive Preference Integration"
_ICLR.cc/2025/Conference — ICLR 2025 Conference Withdrawn Submission_

### Official Review · Reviewer_y6of · 2024-10-27

**Soundness:** 2
**Presentation:** 2
**Contribution:** 2
**Rating:** 3
**Confidence:** 4

**Summary:**

The paper proposes the Preference-Attended Multi-Objective Decision Transformer (PA-MODT) to facilitate the learning across large preference spaces and handling unknown preferences during evaluation. Specifically, a preference-attention block is integrated into the MODT architecture to enhancing preference encoding.

**Strengths:**

The paper provides the insight that promotes a transformer-based model’s performance by modifying the architecture of the transformer.

**Weaknesses:**

- The reference format and the format of the caption of Figure 1 are not consistent with ICLR formatting instructions.
- The discussions on the modification of transformers are better moved to the Related Works section, while also introduce the application of transformers in RL.
- The symbols for vectors are better expressed with bold font in the notation section.
- In Pareto Optimality of Section 3, the explanations of Pareto efficiency is incomplete. Consider a Pareto solution set {[0.25, 0.75], [0.75, 0.25]}, where the two points exceeds the other on one of the objectives, but they are all Pareto efficient, being not consistent with the explanations.
- The ‘key features’ for PAMODT’s construction are proposed without sufficient evidences or reasoning. The novelty and contributions are very limited.
- The expressions of Eq.(1)-(5) are not concise.

**Questions:**

- In line 168, is s_{t+t} a typo? The expression of E in line 191 is better expressed with commas, like E=[p_1, p_2, \ldots, p_m]; In line 198 and the title of Section 5.3, should ‘matrices’ be ‘metrics’ ?
- Are the error bars in Figure 5 lost ? And sparsity metric is better evaluated to provide enough evidences and understandings.
- In Section 5.1 and 5.2, how do the results reflect the property and superiority of the proposed methods ?
- How do the experiments in Section 5.3 relate to the main ideas and contributions of this paper ?

Limitations: The limitations are not discussed in the paper. I think the proposed method has very limited novelty and contributions. Besides, the method’s applicability is also limited since it only suits the transformer-based architectures in MORL.

---

### Official Review · Reviewer_XzsA · 2024-11-04

**Soundness:** 2
**Presentation:** 2
**Contribution:** 2
**Rating:** 3
**Confidence:** 4

**Summary:**

This paper investigates Multi-Objective Reinforcement Learning (MORL). Specifically, authors propose Preference-Attended Multi-Objective Decision Transformer (PA-MODT), a new architecture that introduces a preference-attention block to decision transformer. Experiments on the D4MORL benchmark illustrates the effectiveness of PA-MODT.

**Strengths:**

1. This paper is clearly written and easy to follow.
2. A number of empirical experiments are conducted.

**Weaknesses:**

1. Lack of serious explanations and analysiss on the reasons why the proposed new architecture works better.
2. Taking variance into account, the results shown in Table 1 may be insufficient to demonstrate that PA-MODT significantly outperforms the baseline methods.

**Questions:**

1. In Table 5, the training steps of PA-MODT is not pre-fixed. You claim that evaluation will be conducted every 5k steps. I wonder whether the final evaluation result (presented in Table 1) is the best score selected from all the evaluations. If so, are all the baselines evaluated in this same way?

---

### Official Review · Reviewer_RSH1 · 2024-11-04

**Soundness:** 1
**Presentation:** 2
**Contribution:** 1
**Rating:** 3
**Confidence:** 4

**Summary:**

The paper proposes the Preference-Attended Multi-Objective Decision Transformer (PA-MODT), a new Offline MORL architecture that outperforms existing models on D4MORL.

**Strengths:**

1. The experimental results of PA-MODT are able to surpass the baseline algorithms.
2. The related work is fairly comprehensive.

**Weaknesses:**

Overall, the reviewer believes that this paper falls far short of the quality necessary for acceptance at ICLR. The writing is disorganized and the definitions are unclear, as exemplified by line 283. The main contribution, adding a new attention module to the existing MODT framework, lacks innovation. Additionally, the reviewers cannot understand why the proposed method is effective, as the presentation is very confusing. Finally, the content of the article is insufficient, with the actual method description and explanation taking up less than a page, while the figures are too large and lack information.

**Questions:**

See weakness.

---

### Official Review · Reviewer_rcbG · 2024-11-04

**Soundness:** 3
**Presentation:** 2
**Contribution:** 2
**Rating:** 3
**Confidence:** 4

**Summary:**

The paper presents an empirical analysis on how will the advances in transformer architecture affect the model’s performance on MORL tasks. Specifically, Preference-Attended Multi-Objective Decision Transformer (PA-MODT), a new architecture based on MODT is proposed to facilitate the preference encoding problem.

**Strengths:**

- The paper shed some light on how will the modification of the transformer architecture will affect the performance of MODT models in MORL.
- PA-MODT achieves good performance.

**Weaknesses:**

- There are many typos in the paper.
- The motivations of modifying the transformers’ architecture are not clarified clearly.
- The explanations of Pareto efficiency in Section 3 is incorrect. A point $\beta$ is Pareto efficient if and only if there exists no other point $\alpha$ that behaves no worse than $\beta$ for any objectives, while satisfying $R^\beta_i < R^\alpha_i$ for some $i$.
- The novelty of the proposed method and contributions are very limited.
- Formulas Eq.(1)-(5) should be more formalized.
- The paper does not contain any discussion on limitations. In my opinion, this paper exhibits very limited novelty and applicability, since the methods only fits the transformer-based architectures in MORL, and their properties are not clarified clearly.

**Questions:**

- In line 198 and the title of Section 5.3, should ‘matrices’ be ‘metrics’ ?
- Are the error bars in Figure 5 of Appendix A.1 lost ? Which may lead to unreliable argument in the methods of Section 4.
- In Section 5.1 and 5.2, how do the results reflect the property and superiority of the proposed methods ?
- What do the results in Section 5.1 indicate about the feature or advances of the methods compared to the baselines ?
- What is the relationship between the experiments in Section 5.3 and the proposed methods ?

---

### Note · Authors · 2024-11-15

I have read and agree with the venue's withdrawal policy on behalf of myself and my co-authors.